# Multi-Source Transfer Learning for Deep Model-Based Reinforcement Learning

**Remo Sasso**                                                                         *r.sasso@qmul.ac.uk*
*Queen Mary University of London*

**Matthia Sabatelli**                                                                  *m.sabatelli@rug.nl*
*University of Groningen*

**Marco A. Wiering**                                                                   *m.wiering@rug.nl*
*University of Groningen*

**Reviewed on OpenReview**: *https://openreview.net/forum?id=1nhTDzxxMA*

## Abstract

A crucial challenge in reinforcement learning is to reduce the number of interactions with the environment that an agent requires to master a given task. Transfer learning proposes to address this issue by re-using knowledge from previously learned tasks. However, determining which source task qualifies as the most appropriate for knowledge extraction, as well as the choice regarding which algorithm components to transfer, represent severe obstacles to its application in reinforcement learning. The goal of this paper is to address these issues with modular multi-source transfer learning techniques. The proposed techniques automatically learn how to extract useful information from source tasks, regardless of the difference in state-action space and reward function. We support our claims with extensive and challenging cross-domain experiments for visual control.

## 1 Introduction

Reinforcement learning (RL) offers a powerful framework for decision-making tasks ranging from games to robotics, where agents learn from interactions with an environment to improve their performance over time. The agent observes states and rewards from the environment and acts with a policy that maps states to actions. The ultimate goal of the agent is to find a policy that maximizes the expected cumulative reward corresponding to the task. A crucial challenge in RL is to learn this optimal policy with a limited amount of data, referred to as sample efficiency. This is particularly relevant for real-world applications, where data collection may be costly. In order to achieve sample efficiency, agents must effectively extract information from their interactions with the environment and explore the state space efficiently.

Model-based reinforcement learning methods learn and utilize a model of the environment that can predict the consequences of actions, which can be used for planning and generating synthetic data. Recent works show that by learning behaviors from synthetic data, model-based algorithms exhibit remarkable sample efficiency in high-dimensional environments compared to model-free methods that do not construct a

model (Łukasz Kaiser et al., 2020; Schrittwieser et al., 2020; Hafner et al., 2021). However, a model-based agent needs to learn an accurate model of the environment which still requires a substantial number of interactions. The fact that RL agents typically learn a new task from scratch (i.e. without incorporating prior knowledge) plays a major role in their sample inefficiency. Given the potential benefits of incorporating such prior knowledge (as demonstrated in the supervised learning domain (Yosinski et al., 2014; Sharif Razavian et al., 2014; Zhuang et al., 2020)), transfer learning has recently gained an increasing amount of interest in RL research (Zhu et al., 2020). However, since the performance of transfer learning is highly dependent on the relevance of the data that was initially trained on (source task) with respect to the data that will be faced afterward (target task), its application to the RL domain can be challenging. RL environments namely often differ in several fundamental aspects (e.g. state-action spaces and reward function), making the choice of suitable source-target transfer pair challenging and often based on the intuition of the designer (Taylor & Stone, 2009). Moreover, existing transfer learning techniques are often specifically designed for model-free algorithms (Wan et al., 2020; Heng et al., 2022), transfer within the same domain (Sekar et al., 2020; Liu & Abbeel, 2021; Yuan et al., 2022), or transfer within the same task (Hinton et al., 2015; Parisotto et al., 2015; Czarnecki et al., 2019; Agarwal et al., 2022).

This paper proposes to address the challenge of determining the most effective source task by automating this process with multi-source transfer learning. Specifically, we introduce a model-based transfer learning framework that enables multi-source transfer learning for a state-of-the-art model-based reinforcement learning algorithm across different domains. Rather than manually selecting a single source task, the introduced techniques allow an agent to automatically extract the most relevant information from a *set* of source tasks, regardless of differences between environments. We consider two different multi-source transfer learning settings and provide solutions for each: a single agent that has mastered multiple tasks, and multiple individual agents that each mastered a single task. As modern RL algorithms are often composed of several components we also ensure each of the proposed techniques is applicable in a modular fashion, allowing the applicability to individual components of a given algorithm. The proposed methods are extensively evaluated in challenging cross-domain transfer learning experiments, demonstrating resilience to differences in state-action spaces and reward functions. Note that the techniques are also applicable in single-source transfer learning settings. but we encourage the usage of multi-source transfer learning, in order to avoid the manual selection of an optimal source task. The overall contribution of this paper is the adaptation of existing transfer learning approaches in combination with novel techniques, to achieve enhanced sample efficiency for a model-based algorithm without the need of selecting an optimal source task. The main contributions are summarized as follows:

- **Fractional transfer learning**: We introduce a type of transfer learning that transfers partial parameters instead of random initialization, resulting in substantially improved sample efficiency.

- **Modular multi-task transfer learning**: We demonstrate enhanced sample efficiency by training a single model-based agent on multiple tasks and modularly transferring its components with different transfer learning methods.

- **Meta-model transfer learning**: We propose a multi-source transfer learning technique that combines models from individual agents trained in different environments into a shared feature space, creating a meta-model that utilizes these additional input signals for the target task.

## 2 Preliminaries

**Reinforcement learning**. We formalize an RL problem as a Markov Decision Process (MDP) (Bellman, 1957), which is a tuple $(\mathcal{S}, \mathcal{A}, P, R)$, where $\mathcal{S}$ denotes the state space, $\mathcal{A}$ the action space, $P$ the transition function, and $R$ the reward function. For taking a given action $a \in \mathcal{A}$ in state $s \in \mathcal{S}$, $P(s'|s, a)$ denotes the probability of transitioning into state $s' \in \mathcal{S}$, and $R(r|s, a)$ yields an immediate reward $r$. The state and action spaces define the domain of the MDP. The objective of RL is to find an optimal policy $\pi^* : \mathcal{S} \to \mathcal{A}$ that maximizes the expected cumulative reward. The expected cumulative reward is defined as $G_t = \mathbb{E}\left[\sum_{t=0}^{\infty} \gamma^t R_t\right]$, where $\gamma \in [0, 1)$ represents the discount factor, and $t$ the time-step.

**Transfer learning**. The general concept of transfer learning aims to enhance the learning of some *target* task by re-using information obtained from learning some *source* task. Multi-source transfer learning aims to enhance the learning of a target task by re-using information from a *set* of source tasks. In RL, a task is formalized as an MDP[1] $M$ with some optimal policy $\pi^*$. As such, in multi-source transfer learning for RL we have a collection of $N$ source MDPs $\mathcal{M} = \{(\mathcal{S}_i, \mathcal{A}_i, P_i, R_i)\}_{i=1}^{N}$, each with some optimal policy $\pi_i^*$. Let $M = (\mathcal{S}, \mathcal{A}, P, R)$ denote some target MDP with optimal policy $\pi^*$, where $M$ is different from all $M_i \in \mathcal{M}$ with regards to $\mathcal{S}, \mathcal{A}, P$, or $R$. Multi-source transfer learning for reinforcement learning aims to enhance the learning of $\pi^*$ by reusing information obtained from learning $\pi_1^*, .., \pi_N^*$. When $N = 1$ we have single-source transfer learning.

**Dreamer**. In this paper, we evaluate the proposed techniques on a state-of-the-art deep model-based algorithm, Dreamer (Figure 1, Hafner et al. (2020)). Dreamer constructs a model of the environment within a compact latent space learned from visual observations, referred to as a world model Ha & Schmidhuber (2018). That is, the transition function $P$ and reward function $R$ are modeled with latent representations of environment states. As such, interactions of environments with high-dimensional observations can be simulated in a computationally efficient manner, which facilitates sample-efficient policy learning. The learned policy collects data from the environment, which is then used for learning the world model .

Dreamer consists of six main components:

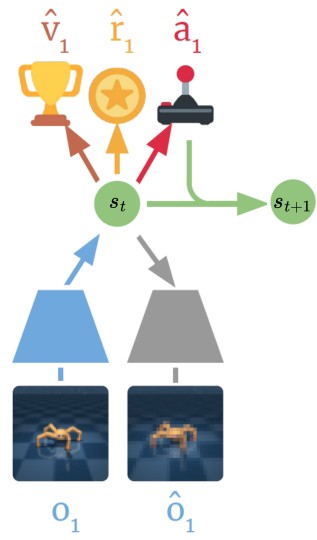

Figure 1: Dreamer maps environment observations $o_t$ to latent states $s_t$ and learns to predict the corresponding reward $r_t$ and next latent state $s_{t+1}$ for a given action $a_t$. These world model components are jointly optimized with a reconstruction loss from the observation model. The world model is then used to simulate environment trajectories for learning an actor-critic policy, which predicts actions $a_t$ and value estimates $v_t$ that maximize future value predictions. Arrows represent the parameters of a given model.

- **Representation model**:    $p_{\theta_{\text{REP}}}(s_t|s_{t-1}, a_{t-1}, o_t)$.

- **Observation model**:    $q_{\theta_{\text{OBS}}}(o_t|s_t)$.

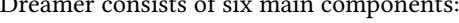

[1]We use the terms MDP, task, and environment interchangeably in this paper.

- **Reward model:** $\qquad$ $q_{\theta_{\text{REW}}}(r_t|s_t)$.

- **Transition model:** $\qquad$ $q_{\theta_{\text{TRANS}}}(s_t|s_{t-1}, a_{t-1})$.

- **Actor:** $\qquad$ $q_\phi(a_\tau|s_\tau)$.

- **Critic:** $\qquad$ $v_\psi(s_t) \approx \mathbb{E}_{q(\cdot|s_\tau)}(\sum_{\tau=t}^{t+H} \gamma^{\tau-t} r_\tau)$.

Here $p$ denotes distributions that generate real environment samples, $q$ denotes distributions approximating those distributions in latent space, and $\theta$ denotes the jointly optimized parameters of the models. At a given timestep $t$, the representation model maps a visual observation $o_t$ together with the previous latent state $s_{t-1}$ and previous action $a_{t-1}$ to latent state $s_t$. The transition model learns to predict $s_{t+1}$ from $s_t$ and $a_t$, and the reward model learns to predict the corresponding reward $r_t$. The observation model reconstructs $s_t$ to match $o_t$, providing the learning signal for learning the feature space. This world model is called the recurrent state space model (RSSM), and we refer the reader to Hafner et al. (2019) for further details. To simulate a trajectory $\{(s_\tau, a_\tau)\}_{\tau=t}^{t+H}$ of length $H$, where $\tau$ denotes the imagined time index, the representation model maps an initial observation $o_t$ to latent state $s_\tau$, which is combined with action $a_\tau$ yielded by the policy, to predict $s_{\tau+1}$ using the transition model. The reward model then predicts the corresponding reward $r_{\tau+1}$, which is used for policy learning.

In order to learn a policy, Dreamer makes use of an actor-critic approach, where the action model $q_\phi(a_\tau|s_\tau)$ implements the policy, and the value model $v_\psi(s_t) \approx \mathbb{E}_{q(\cdot|s_\tau)}(\sum_{\tau=t}^{t+H} \gamma^{\tau-t} r_\tau)$ estimates the expected reward that the action model achieves from a given state $s_\tau$. Here $\phi$ and $\psi$ are neural network parameters for the action and value model respectively, and $\gamma$ is the discount factor. The reward, value, and actor models are implemented as Gaussian distributions parameterized by feed-forward neural networks. The transition model is a Gaussian distribution parameterized by a Gated Recurrent United (GRU; Bahdanau et al. (2014)) followed by feed-forward layers. The representation model is a variational encoder (Kingma & Welling, 2013; Rezende et al., 2014) combined with the GRU, followed by feed-forward layers. The observation model is a transposed convolutional neural network (CNN; LeCun et al. (2015)). See Appendix F for further details and pseudocode of the Dreamer algorithm.

## 3 Related Work

**Transfer learning**. In supervised learning, parameters of a neural network are transferred by either freezing or retraining the parameters of the feature extraction layers, and by randomly initializing the parameters of the output layer to allow adaptation to the new task (Yosinski et al., 2014; Sabatelli et al., 2018; Cao et al., 2021). Similarly, we can transfer policy and value models, depending on the differences of the state-action spaces, and reward functions between environments (Carroll & Peterson, 2002; Schaal et al., 2004; Fernández & Veloso, 2006; Rusu et al., 2016; Zhang et al., 2018). We can also transfer autoencoders, trained to map observations to latent states (Chen et al., 2021). Experience samples collected during the source task training process can also be transferred to enhance the learning of the target task (Lazaric et al., 2008; Tirinzoni et al., 2018).

**Model-free transfer learning**. Model-free reinforcement learning algorithms seem particularly suitable for distillation techniques and transfer within the same domain and task (Hinton et al., 2015; Parisotto et al., 2015; Czarnecki et al., 2019; Agarwal et al., 2022). By sharing a distilled policy across multiple agents learning individual tasks, Teh et al. (2017) obtain robust sample efficiency gains. Sabatelli & Geurts (2021)

showed that cross-domain transfer learning in the Atari benchmark seems to be a less promising application of deep model-free algorithms. More impressive learning improvements are observed in continuous control settings for transfer within the same domain or transfer across fundamentally similar domains with single and multiple policy transfers (Wan et al., 2020; Heng et al., 2022). Our techniques are instead applicable to model-based algorithms and evaluated in challenging cross-domain settings with fundamentally different environments. Ammar et al. (2014) introduce an autonomous similarity measure for MDPs based on restricted Boltzmann machines for selecting an optimal source task, assuming the MDPs are within the same domain. García-Ramírez et al. (2021) propose to select the best source models among multiple model-free models using a regressor. In multiple-source policy transfer, researchers address similar issues but focus on transferring a set of source policies with mismatching dynamics to some target policy in model-free settings (Yang et al., 2020; Barekatain et al., 2020; Lee et al., 2022).

**Model-based transfer learning**. In model-based reinforcement learning one also needs to transfer a dynamics model, which can be fully transferred between different domains if the environments are sufficiently similar (Eysenbach et al., 2021; Rafailov et al., 2021). Landolfi et al. (2019) perform a multi-task experiment, where the dynamics model of a model-based agent is transferred to several novel tasks sequentially, and show that this results in significant gains of sample efficiency. PlaNet (Hafner et al., 2019) was used in a multi-task experiment, where the same agent was trained on tasks of six different domains simultaneously using a single world model (Ha & Schmidhuber, 2018). A popular area of research for transfer learning in model-based reinforcement learning is task-agnostically pre-training the dynamics model followed and introducing a reward function for a downstream task afterward (Rajeswar et al., 2022; Yuan et al., 2022). In particular, in Plan2Explore (Sekar et al., 2020) world model is task-agnostically pre-trained, and a Dreamer (Hafner et al., 2020) agent uses the model to zero-shot or few-shot learn a new task within the same domain. Note that Dreamer is not to be confused with DreamerV2 (Hafner et al., 2021), which is essentially the same algorithm adapted for discrete domains. Unlike previous works, we investigate multi-source transfer learning with a focus on such world model-based algorithms, by introducing techniques that enhance sample efficiency without selecting a single source task, are applicable across domains, are pre-trained with reward functions, and allow the transfer of each individual component.

## 4 Methods

Here we present the main contributions of this work and describe their technical details. First, we introduce multi-task learning as a multi-source setting and combine it with different types of transfer learning in a modular fashion, after introducing a new type of transfer learning that allows for portions of information to be transferred (Section 4.1). We follow this with an alternative setting, where we propose to transfer components of multiple individual agents utilizing a shared feature space and a meta-model instead (Section 4.2).

### 4.1 Multi-Task Transfer Learning

In this section, we describe how a single agent can be trained on multiple MDPs simultaneously, introduce a novel type of transfer learning, and provide insights on transfer learning in a modular manner. These three concepts are combined to create a modular multi-source transfer learning technique that can autonomously learn to extract relevant information for a given target task.

**Simultaneous multi-task learning**. We propose to train a single agent facing multiple unknown environments simultaneously, each with different state-action spaces and reward functions. Intuitively, the

parameters of this agent will contain aggregated information from several source tasks, which can then be transferred to a target task (similar to parameter fusion in the supervised learning domain (Kendall et al., 2018)). We train the multi-task RL agents by padding the action space of the agent with unused elements to the size of the task with the largest action dimension. Note that we are required to have uniform action dimensions across all source environments, as we use a single policy model. The agent collects one episode of each task in collection phases to acquire a balanced amount of experiences in the replay buffer.

**Fractional transfer learning**. When two tasks are related but not identical, transferring all parameters of a neural network may not be beneficial, as it can prevent the network from adapting and continuing to learn the new task. Therefore, the feature extraction layers of a network are typically fully transferred, but the output layer is randomly initialized. We propose a simple alternative called *fractional transfer learning* (FTL). By only transferring a fraction of the parameters, the network can both benefit from the transferred knowledge and continue to learn and adapt to the new task. Let $\theta_T$ denote target parameters, $\theta_\epsilon$ randomly initialized weights, $\lambda$ the fraction parameter, and $\theta_S$ source parameters, then we apply FTL by $\theta_T \leftarrow \theta_\epsilon + \lambda\theta_S$. That is, we add a fraction of the source parameters to the randomly initialized weights. This approach is similar to the shrink and perturb approach presented in supervised learning, where Gaussian noise is added to shrunken weights from the previous training iteration (Ash & Adams, 2020). However, we present a reformulation and simplification where we omit the 'noise scale' parameter $\sigma$ and simply use Xavier uniform weight initialization Glorot et al. (2011) as the perturbation, such that we only need to specify $\lambda$. In addition to recent applications to the model-free setting (Liu et al., 2019; Shu et al., 2021), we are the first to demonstrate the application of this type of transfer learning in a deep model-based RL application.

**Modular transfer learning**. Modern RL algorithm architectures often consist of several components, each relating to different elements of an MDP. Therefore, to transfer the parameters of such architectures, we need to consider transfer learning on a modular level. Using Dreamer as a reference, we discuss what type of transfer learning each component benefits most from. We consider three types of direct transfer learning (i.e. initialization techniques): random initialization, FTL, and full transfer learning, where the latter means we initialize the target parameters with the source parameters. As commonly done in transfer learning, we fully transfer feature extraction layers of each component and only consider the

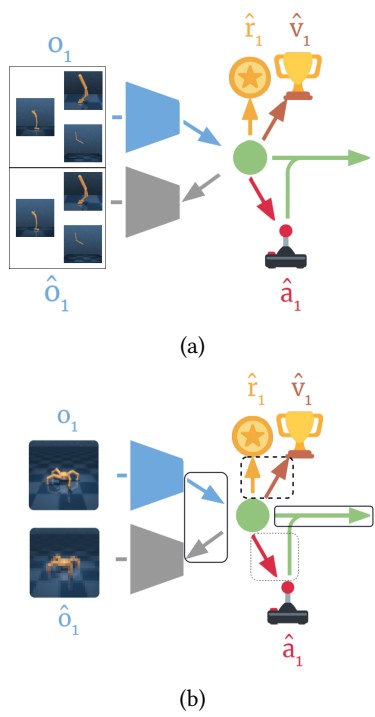

Figure 2: (a) We train a single agent in multiple (source) environments simultaneously by alternately interacting with each environment. (b) We transfer the parameters of this agent in a modular fashion for learning a novel (target) task. The representation, observation and transition model are fully transferred (——). The reward and value models are fractionally transferred (- - -). The action model and the action-input parameters of the transition model are randomly initialized (- - -).

output layer for alternative transfer learning strategies. We don't discuss the action parameters, as ac-

tion elements and dimensions don't match across different environments, meaning we simply randomly initialize action parameters.

We found that fractionally transferring parameters of the reward and value models can result in substantial performance gains (Appendix E). In this paper, we apply our methods to MDPs with similar reward functions, meaning the parameters of these models consist of transferable information that enhances the learning of a target task. This demonstrates the major benefits of FTL, as the experiments also showed that fully transferring these parameters has a detrimental effect on learning (Appendix D). From the literature, we know that fully transferring the parameters of the representation, observation, and transition model often results in positive transfers. As we are dealing with visually similar environments, the generality of convolutional features allows the full transfer of the representation and observation parameters (Chen et al., 2021). Additionally, when environments share similar physical laws, transition models may be fully transferred, provided that the weights connected to actions are reset (Landolfi et al., 2019). See Appendix F for pseudocode and implementation details of the proposed modular multi-task transfer learning approach, and see Figure 2 for an overview of our approach.

## 4.2 Multiple-Agent Transfer Learning

We now consider an alternative multi-source transfer learning setting, where we have access to the parameters of multiple individual agents that have learned the optimal policy for different tasks. Transferring components from agents trained in different environments represents a major obstacle for multi-source transfer learning. To the best of our knowledge, we are the first to propose a solution that allows the combination of deep model-based reinforcement learning components trained in different domains, from which the most relevant information can autonomously be extracted for a given target task.

**Universal feature space**. As the focus of this paper is on world model-based agents that learn their components in a compact latent feature space, we are required to facilitate a shared feature space across the agents when combining and transferring their components. From the literature, we know that for visually similar domains, an encoder of a converged autoencoder can be reused due to the generality of convolutional features (Chen et al., 2021). Based on this intuition we introduce *universal feature space* (UFS): a fixed latent feature space that is shared among several world model-based agents to enable the transferability of their components. We propose to train a single agent in multiple environments simultaneously (as described in Section 4.1), freeze its encoder, and reuse it for training both source and target agents. In this paper, we use two different types of environments: locomotion and pendula tasks (Figure 4). Therefore, we decided to train a single agent simultaneously in one locomotion and one pendulum environment, such that we learn convolutional features for both types of environments. Note that in the case of Dreamer, we do not transfer and freeze the other RSSM components, as this would prevent the Dreamer agent from learning new reward and transition functions. Dreamer learns its latent space via reconstruction, meaning the encoder is the main component of the representation model responsible for constructing the feature space.

**Meta-model transfer learning**. When an agent uses the UFS for training in a given target task, we can combine and transfer components from agents that were trained using the UFS in different environments. We propose to accomplish this by introducing *meta-model transfer learning* (MMTL). For a given component of the target agent, we assemble the same component from all source agents into an ensemble of frozen components. In addition to the usual input element(s), the target component also receives the output signals provided by the frozen source components.

Let $m_\Theta(y|x)$ denote a neural network component with parameters $\Theta$, some input $x$, and some output $y$, belonging to the agent that will be trained on target MDP $M$. Let $m_{\theta_i}(y|x)$ denote the same component belonging to some other agent $i$ with *frozen* parameters $\theta_i$ that were fit to some source MDP $M_i$, where $i \in N$, and $N$ denotes the number source MDPs on which a separate agent was trained on from the set of source MDPs $\mathcal{M}$. In MMTL, we modify $m_\Theta(y|x)$, such that we get:

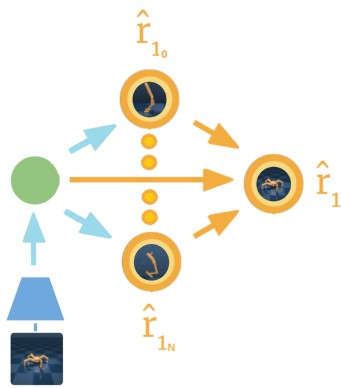

$$m_\Theta(\, y \mid x,\ m_{\theta_0}(y|x),\ ...,\ m_{\theta_N}(y|x)\,)$$

where all models were trained within the same UFS. Intuitively, information signals provided by the source models can be used to enhance the learning process of $m_\Theta$. For instance, assume the objective of $M_i$ is similar to the objective in $M$. In that case, if we use MMTL for the reward model, $m_\Theta$ can autonomously learn to utilize the predictions of $m_{\theta_{\text{REW},i}}$ via gradient descent. Similarly, it can learn to ignore the predictions of source models that provide irrelevant predictions. As we are dealing with locomotion and pendula environments in this paper, we choose to apply MMTL to the reward model of Dreamer (Figure 3), as the locomotion MDPs share a similar objective among each other whilst the pendula environments will provide irrelevant signals for the locomotion environments and vice versa. As such, we can evaluate whether our approach can autonomously learn to utilize and ignore relevant and irrelevant information signals respectively. Note that in the case of Dreamer's reward model, $y$ corresponds to a scalar Gaussian parameterized by a mean $\mu$ and unit variance, from which the mode is sampled. Hence, in our case, $y$ corresponds to $\mu$, and $x$ corresponds to some latent state $s$. As such, the target agent will use the same frozen encoder used by the source agents, in addition to using a reward meta-model. We don't transfer any other components of the architecture in order to be able to observe the isolated effect of the reward meta-model.

Figure 3: Illustration of meta-model transfer learning for Dreamer's reward model. The frozen parameters (cyan) of the encoder provide a fixed mapping of environment observations to latent states, which was also used in training each of the source reward models. This latent state is fed to both the source reward models and the target reward model, of which the latter also takes the predictions of the source reward models as input.

This approach applies the same principles as Progressive Neural Networks (Rusu et al., 2016), as we are retaining, freezing, and combining models from previous learning tasks to enhance the performance of novel tasks. Therefore, MMTL is essentially an application of Progressive Neural Networks in a multi-source and world model-based setting, where we are dealing with transferring components of multiple agents within the latent space of an autoencoder. By leveraging UFS. we enable this combination and transfer of components trained in different domains. Note that one drawback of Progressive Neural Networks is that the computation scales linearly with the number of source models, which also applies in our case with the number of source tasks. The frozen autoencoder provides a slight compensation for this additional computation, as it is no longer updated in the MMTL approach. See Appendix F for pseudocode and implementation details of MMTL.

## 5 Experiments

**Experimental setup**. We evaluate the proposed methods using Dreamer[2] on six continuous control tasks: Hopper, Ant, Walker2D, HalfCheetah, InvertedPendulumSwingup, and InvertedDoublePendulumSwingup

---

[2]This work builds upon the code base of Dreamer: `https://github.com/danijar/dreamer`.

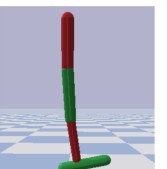 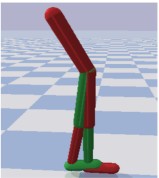 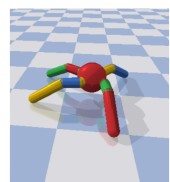 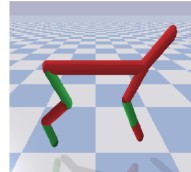 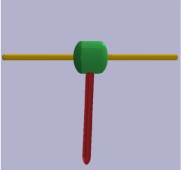 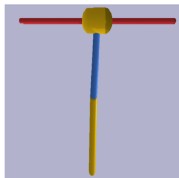

Figure 4: Visualization of the tasks learned in the experiments: Hopper, Walker2D, Ant, HalfCheetah, InvertedPendulumSwingup, and InvertedDoublePendulumSwingup. Each of the MDPs differ in all elements $(\mathcal{S}, \mathcal{A}, R, P)$, and are used in multi-source transfer learning experiments as both source and target tasks.

Table 1: Overall average episode return of 1 million environment steps for FTL and MMTL, where parameters of 4 source tasks were transferred. We compare to a baseline Dreamer agent that learns from scratch. Bold results indicate the method outperforms the baseline for a given task.

| Task | FTL | MMTL | Baseline |
|------|-----|------|----------|
| HalfCheetah | **1490 ± 441** | **1882 ± 390** | 1199 ± 558 |
| Hopper | **3430 ± 2654** | **4025 ± 3404** | 2076 ± 2417 |
| Walker2D | **1637 ± 2047** | **847 ± 1533** | 676 ± 1101 |
| InvPend | **761 ± 68** | **705 ± 79** | 667 ± 121 |
| InvDbPend | **1235 ± 130** | **1198 ± 100** | 1184 ± 89 |
| Ant | 681 ± 591 | 1147 ± 922 | 1148 ± 408 |

(Figure 4). Due to a restricted computational budget, we provide extensive empirical evidence for only one model-based algorithm and leave the investigation of the potential benefits for other (model-based) algorithms to future work. A detailed description of the differences between the MDPs can be found in Appendix A. We perform experiments for both multi-task (referred to as FTL) and multiple-agent (referred to as MMTL) transfer learning settings. For each method, we run multi-source transfer learning experiments using a different set of $N$ source tasks for each of the target environments (Appendix B). The selection of source tasks for a given set was done such that each source set for a given target environment includes at least one task from a different environment type, i.e., a pendulum task for a locomotion task and vice versa. Similarly, each source set contains at least one task of the same environment type. We also ran preliminary experiments (3 random seeds) for sets consisting of $N = [2, 3]$ to observe potential performance differences that result from different $N$, but we found no significant differences (Appendix C).

**Hyperparameters**. To demonstrate that our methods can autonomously extract useful knowledge from a set of source tasks that includes at least one irrelevant source task, we apply our methods to the most challenging setting ($N = 4$) for 9 random seeds. To the best of our knowledge there exist no available and comparable multi-source transfer learning techniques in the literature that are applicable to cross-domain transfer learning or applicable on a modular level to a world model-based algorithm such as Dreamer. Therefore, for each run, we train the FTL and MMTL target agents for 1 million environment steps and compare them to a baseline Dreamer agent that learns from scratch for 1 million environment steps, in order to evaluate the sample efficiency gains of the transfer learning approaches. FTL is evaluated by training multi-task agents for 2 million environment steps for a single run, after which we transfer the parameters to the target agent as described in Section 4.1. We use a fraction of $\lambda = 0.2$ for FTL, as we observed in preliminary experiments, the largest performance gains occur in a range of $\lambda \in [0.1, 0.3]$ (see Appendix E).

Table 2: Average episode return of the final 1e5 environment steps for FTL and MMTL, where parameters of 4 source tasks were transferred. We compare to a baseline Dreamer agent that learns from scratch. Bold results indicate the method outperforms the baseline for a given task.

| Task | FTL | MMTL | Baseline |
|---|---|---|---|
| HalfCheetah | **2234 ± 302** | **2458 ± 320** | 1733 ± 606 |
| Hopper | **5517 ± 4392** | **7438 ± 4157** | 3275 ± 3499 |
| Walker2D | **2750 ± 2702** | **1686 ± 2329** | 1669 ± 1862 |
| InvPend | **879 ± 17** | 872 ± 20 | 875 ± 23 |
| InvDbPend | **1482 ± 162** | 1370 ± 106 | 1392 ± 115 |
| Ant | 1453 ± 591 | 1811 ± 854 | 1901 ± 480 |

For creating the UFS for MMTL, we train a multi-task agent on the Hopper and InvertedPendulum task for 2 million environment steps. We evaluate MMTL by training a single agent on each task of the set of source environments for 1 million environment steps (all using the same UFS), after which we transfer their reward models to the target agent as described in Section 4.2. We used a single Nvidia V100 GPU for each training run, taking about 6 hours per 1 million environment steps.

**Results**. The overall aggregated return of both FTL and MMTL is reported in Table 1, which allows us to take jumpstarts into account in the results. The aggregated return of the final 10% (1e5) environment steps can be found in Table 2, allowing us to observe final performance improvements. Figure 5 shows the corresponding learning curves.

## 6 Discussion

We now discuss the results of each of the proposed methods individually and reflect on the overall multi-source transfer learning contributions of this paper. We would like to emphasize that we don't compare FTL and MMTL to each other, but to the baseline, as they are used for two entirely different settings. Additionally, although the proposed techniques are applicable to single-source transfer learning settings as well, empirical performance comparisons between multi-source and single-source transfer learning are outside of the scope of this paper, and we leave this to future work.

**Multi-source transfer learning**. The overall transfer learning results show that the proposed solutions for the multi-task setting (FTL) and multiple-agent setting (MMTL) result in positive transfers for 5 out of 6 environments. We observe jumpstarts, overall performance improvements, and/or final performance gains. This shows the introduced techniques allow agents to autonomously extract useful information stemming from source agents trained in MDPs with different state-action spaces, reward functions, and transition functions. We would like to emphasize that for each of the environments, there were at most two environments that could provide useful transfer knowledge. Nevertheless, our methods are still able to identify the useful information and result in a positive transfer with 4 source tasks. For the Ant testing environment, we observe negative transfers when directly transferring parameters in the multi-task setting, which logically follows from the environment being too different from all other environments in terms of dynamics, as we fully transfer the transition model. Note that MMTL does not suffer significantly from this difference in environments, as the transition model is not transferred in that setting.

**Multi-task and fractional transfer learning**. When training a single agent on multiple tasks and transferring its parameters to a target task in a fused and modular fashion, and applying FTL to the reward and

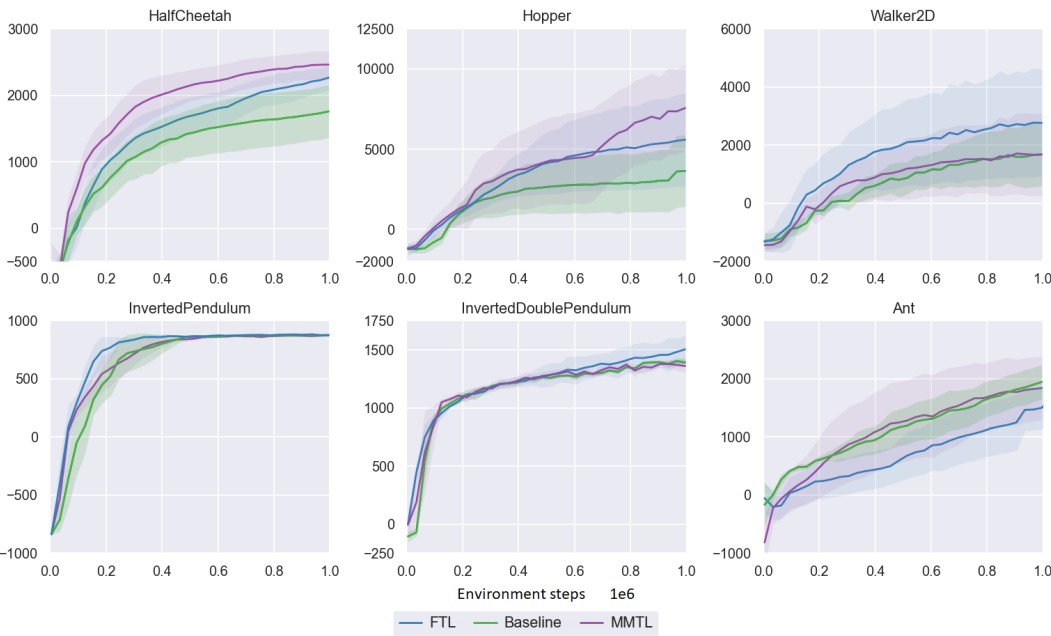

Figure 5: Learning curves for average episode return of 1 million environment steps for FTL and MMTL using 4 source tasks, compared to a baseline Dreamer agent that was trained from scratch. Shaded areas represent standard error across the 9 runs.

value models, we observe impressive performance improvements in 5 out of 6 environments. In particular, consistent performance improvements are obtained for the three locomotion tasks that can be considered related, suggesting that fusing parameters of multiple tasks alleviates having to choose a single optimal source task for transfer learning. This is further demonstrated in the results of the pendula environments, where just one out of four source tasks is related to the target task. Regardless, consistent and significant jumpstarts are obtained compared to training from scratch (see Figure 5). Instead of discarding parameters of output layers, and instead transferring fractions of their parameters can play a major role in such performance improvements, as initial experiments demonstrated (Appendix E). However, one drawback of FTL is that it introduces a tunable fraction parameter $\lambda$, where the optimal fraction can differ per environment and composition of source tasks. We observed the most performance gains in a range of $\lambda \in [0.1, 0.3]$, and for larger fractions, the overall transfer learning performance of the agent would degrade. Similarly, Ash & Adams (2020) found that $\lambda = 0.2$ provided the best performance in the supervised learning domain for the shrink and perturb method.

**Meta-model transfer learning.** An alternative multi-source setting we considered is where we aim to transfer information of multiple individual agents, each trained in different environments. By training both source and target agents with a shared feature space and combining their components into a meta-model, we observe substantial performance gains when merely doing so for the reward model. The results suggest that the target model learns to leverage the additional learning signals provided by the frozen reward models trained in the source tasks. In particular, we observe impressive performance improvements in locomotion environments, which are visually similar and share identical objectives. This suggests that the

UFS enables the related source reward models to provide useful predictions for an environment they were not trained with. Note that the source models are also not further adapted to the new environment as their parameters are frozen. The overall performance improvements are less substantial, yet still present for the pendula environments. As initial experiments suggested, there being just one related source task for the pendula environments leads to degrading learning performance when adding additional unrelated source models (Appendix C). To the best of our knowledge, we are the first to successfully combine individual components of multiple individual agents trained in different environments as a multi-source transfer learning technique that results in positive transfers, which is autonomously accomplished through gradient descent and a shared feature space provided by the proposed UFS approach. Note that even though there is additional inference of several frozen reward models, for MMTL there is little additional computational complexity as the frozen autoencoder does not require gradient updates, compensating for any additional computation.

## 7 Conclusion

In this paper, we introduce several techniques that enable the application of multi-source transfer learning to modern model-based algorithms, accomplished by adapting and combining both novel and existing concepts from the supervised learning and reinforcement learning domains. The proposed methods address two major obstacles in applying transfer learning to the RL domain: selecting an optimal source task and deciding what type of transfer learning to apply to individual components. The introduced techniques are applicable to several individual deep reinforcement learning components and allow the automatic extraction of relevant information from a set of source tasks. Moreover, the proposed methods cover two likely cross-domain multi-source scenarios: transferring parameters from a single agent that mastered several tasks and transferring parameters from multiple agents that mastered a single task. Where existing techniques on transfer learning generally focus on model-free algorithms, transfer within the same domain, or transfer within the same task, we showed that our introduced methods are applicable to challenging cross-domain settings and compatible with a state-of-the-art model-based reinforcement learning algorithm, importantly without having to select an optimal source task.

First, we introduced fractional transfer learning, which allows parameters to be transferred partially, as opposed to discarding information as commonly done by randomly initializing a layer of a neural network. We used this type of transfer learning as an option in an insightful discussion concerning what type of transfer learning deep model-based reinforcement learning algorithm components benefit from. The conclusions of this discussion were empirically validated in the multi-task transfer learning setting, which both showed that the modular transfer learning decisions result in significant performance improvements for learning a novel target task, and that fused parameter transfer allows for autonomous extraction of useful information from multiple different source tasks.

Next, we showed that by learning a universal feature space, we enable the combination and transfer of individual components from agents trained in environments with different state-action spaces and reward functions. We extend this concept by introducing meta-model transfer learning, which leverages the predictions of models trained by different agents in addition to the usual input signals, as a multi-source transfer learning technique for multiple-agent settings. This again results in significant sample efficiency gains in challenging cross-domain experiments while autonomously learning to leverage and ignore relevant and irrelevant information, respectively.

A natural extension for future work is the application of these techniques beyond the experimental setup used in this research, such as different environments (e.g. Atari 2600) and algorithms. The techniques introduced in this paper are applicable on a modular level, indicating potential applicability to components of other model-based algorithms, and potentially model-free algorithms as well. Another interesting direction would be the application of the proposed techniques in a single-source fashion, and empirically comparing the transfer learning performance to the multi-source approach.

**Acknowledgments**

We thank the Center for Information Technology of the University of Groningen for their support and for providing access to the Peregrine high-performance computing cluster.

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

## A    Environment Descriptions

In this paper locomotion and pendulum balancing environments from PyBullet (Coumans & Bai, 2016–2021) are used for experiments. The locomotion environments have as goal to walk to a target point that is distanced 1 kilometer away from the starting position as quickly as possible. Each environment has a different entity with different numbers of limbs and therefore has different state-action spaces, and transition functions. The reward function is similar, slightly adapted for each entity as the agent is penalized for certain limbs colliding with the ground. The pendula environments have as their objective to balance the initially inverted pendulum upwards. The difference between the two environments used is that one environment has two pendula attached to each other. This environment is not included in the PyBullet framework for swing-up balancing, which we, therefore, implemented ourselves. The reward signal for the InvertedPendulumSwingup for a given observation $o$ is:

$$r_o = \cos \Theta \tag{1}$$

where $\Theta$ is the current position of the joint. For the InvertedDoublePendulumSwingup a swing-up task, we simply add the cosine of the position of the second joint $\Gamma$ to Equation 1:

$$r_f = \cos \Theta + \cos \Gamma \tag{2}$$

As such, these two environments also differ in state-action spaces, transition functions, and reward functions.

## B    Transfer Learning Task Combinations

In Table 3 the combinations of source tasks and target tasks can be viewed that were used for the experiments in this paper. That is, they correspond to the results of Appendix C and Section 5.

Table 3: Source-target combinations used for FTL and MMTL experiments, using environments HalfCheetah (Cheetah), Hopper, Walker2D, InvertedPendulumSwingup (InvPend), InvertedDoublePendulumSwingup (InvDbPend), and Ant.

| Target | 2 Tasks | 3 Tasks | 4 Tasks |
|---|---|---|---|
| Cheetah | Hopper, Ant | +Walker2D | +InvPend |
| Hopper | Cheetah, Walker2D | +Ant | +InvPend |
| Walker2D | Cheetah, Hopper | +Ant | +InvPend |
| InvPend | Cheetah, InvDbPend | +Hopper | +Ant |
| InvDbPend | Hopper, InvPend | +Walker2D | +Ant |
| Ant | Cheetah, Walker2D | + Hopper | +InvPend |

## C   Preliminary Experiments for 2 and 3 Source Tasks

In Table 4 and Table 5 the results for training on, and transferring, 2 and 3 source tasks as described in Section 5 can be found for both FTL and MMTL, showing the overall average episode return and the episode return for the final 1e5 environment steps respectively. In Figure 6 the corresponding learning curves can be found. These experiments are the averages and standard deviation across 3 seeds for FTL, MMTL, and a baseline Dreamer trained from scratch. As just 3 seeds were used for these experiments, they are not conclusive. However, they do show that regardless of the number of source tasks, our methods can extract useful information and enhance the performance compared to the baseline.

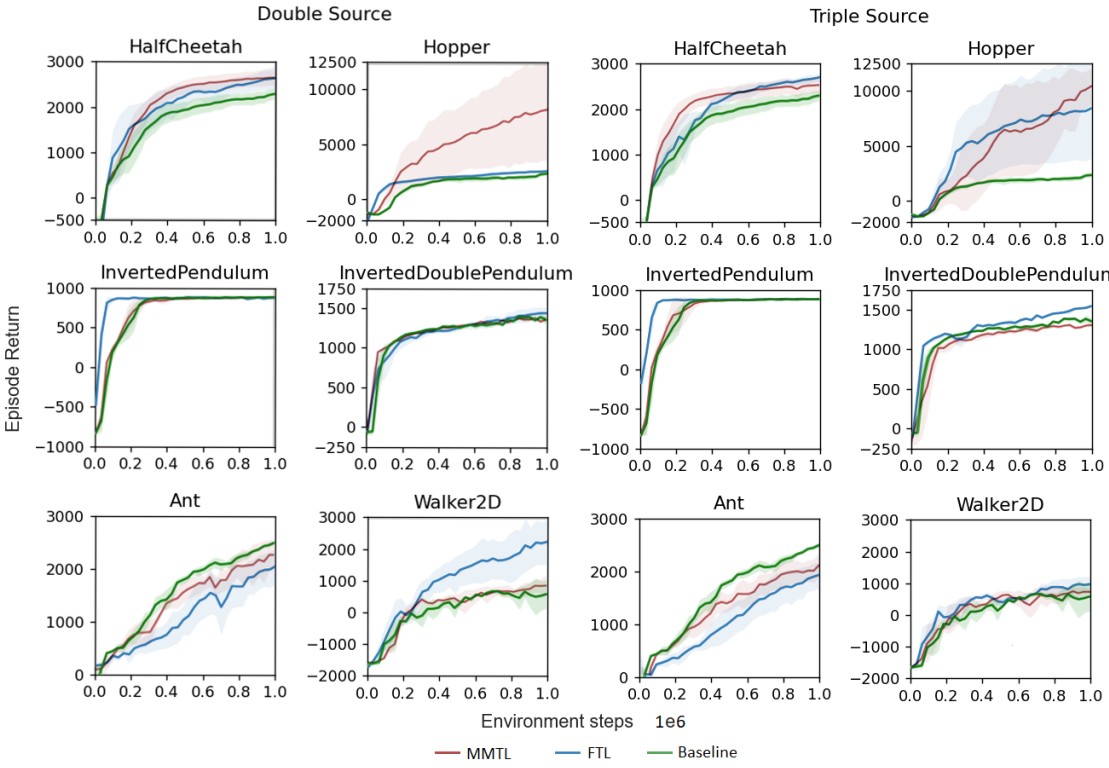

Figure 6: Preliminary experiments: average episode return for 3 random seeds obtained across 1 million environment steps by MMTL (red), FTL (blue), and a vanilla Dreamer (green), where for FTL $\lambda = 0.2$ is used. FTL and MMTL both receive transfer from a combination of 2 (double) and 3 (triple) source tasks. Shaded areas represent the standard deviation across the 3 runs.

Table 4: Preliminary experiments: overall average episode return with of FTL using $\lambda$ = 0.2 and MMTL for the reward model. Parameters of 2 and 3 source tasks are transferred to the HalfCheetah, Hopper, Walker2D, InvertedPendulumSwingup (InvPend), InvertedDoublePendulumSwingup (InvDbPend), and Ant tasks, and compared to a baseline Dreamer agent that learns from scratch. Bold results indicate the best performance across the methods and baseline for a given task.

|  | Fractional Transfer Learning | | Meta-Model Transfer Learning | | |
| Task | 2 task | 3 task | 2 task | 3 task | Baseline |
| --- | --- | --- | --- | --- | --- |
| HalfCheetah | 1982 ± 838 | 1967 ± 862 | 2057 ± 851 | 2078 ± 721 | 1681 ± 726 |
| Hopper | 1911 ± 712 | 5538 ± 4720 | 5085 ± 4277 | 5019 ± 4813 | 1340 ± 1112 |
| Walker2D | 1009 ± 1254 | 393 ± 813 | 196 ± 860 | 233 ± 823 | 116 ± 885 |
| InvPend | 874 ± 121 | 884 ± 20 | 731 ± 332 | 740 ± 333 | 723 ± 364 |
| InvDbPend | 1209 ± 280 | 1299 ± 254 | 1214 ± 230 | 1120 ± 327 | 1194 ± 306 |
| Ant | 1124 ± 722 | 1052 ± 687 | 1423 ± 788 | 1366 ± 687 | 1589 ± 771 |

Table 5: Preliminary experiments: average episode return of the final 1e5 environment steps of FTL using $\lambda$ = 0.2 and MMTL for the reward model. Parameters of 2 and 3 source tasks are transferred to the HalfCheetah, Hopper, Walker2D, InvertedPendulumSwingup (InvPend), InvertedDoublePendulum-Swingup (InvDbPend), and Ant tasks, and compared to a baseline Dreamer agent that learns from scratch. Bold results indicate the best performance across the methods and baseline for a given task.

|  | Fractional Transfer Learning | | Meta-Model Transfer Learning | | |
| Task | 2 task | 3 task | 2 task | 3 task | Baseline |
| --- | --- | --- | --- | --- | --- |
| HalfCheetah | 2820 ± 297 | 2615 ± 132 | 2618 ± 362 | 2514 ± 183 | 2264 ± 160 |
| Hopper | 2535 ± 712 | 8274 ± 46 490 | 8080 ± 4704 | 10 178 ± 2241 | 2241 ± 502 |
| Walker2D | 2214 ± 1254 | 963 ± 314 | 846 ± 286 | 730 ± 343 | 547 ± 710 |
| InvPend | 874 ± 121 | 884 ± 20 | 885 ± 14 | 884 ± 13 | 883 ± 17 |
| InvDbPend | 1438 ± 116 | 1531 ± 93 | 1348 ± 171 | 1302 ± 152 | 1366 ± 179 |
| Ant | 2021 ± 722 | 1899 ± 292 | 2212 ± 607 | 2064 ± 386 | 2463 ± 208 |

## D  Full Transfer

In Figure 7 learning curves can be seen for the HalfCheetah task where we fully transfer the parameters instead of using FTL. As can be seen, this results in a detrimental overfitting effect where the agent does not seem to be learning.

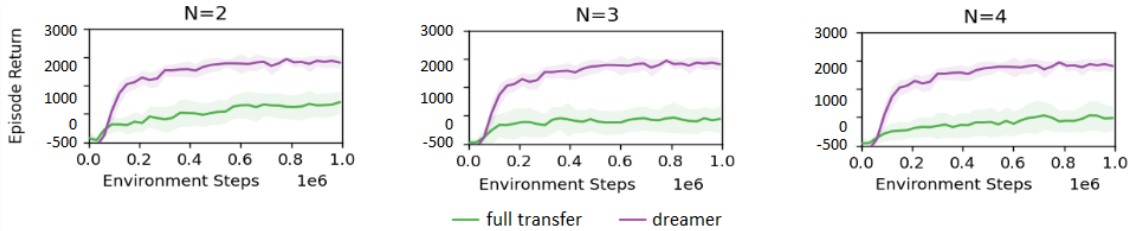

Figure 7: 4 seeds for full transfer learning on the HalfCheetah task across 2, 3, and 4 source tasks.

## E  Additional Fraction Results

In this appendix, results can be found that illustrate the effect of different fractions $\lambda$. We present the numerical results of transferring fractions of $\lambda \in [0.0, 0.1, 0.2, 0.3, 0.4, 0.5]$ using 2, 3, and 4 source tasks. That is, the overall performance over 3 random seeds for the HalfCheetah and Hopper, Walker2D and Ant, and InvertedPendulumSwingup and InvertedDoublePendulumSwingup testing environments can be found in Table 6, Table 7, and Table 8 respectively. Again, as just 3 seeds were used for these experiments, they are not conclusive. However, they do indicate that the choice of $\lambda$ can largely impact performance, and generally, the best performance gains are yielded with $\lambda \in [0.1, 0.3]$.

Table 6: Average return for fraction transfer of 2, 3, and 4 source tasks for the HalfCheetah and Hopper tasks, with fractions $\lambda \in [0.0, 0.1, 0.2, 0.3, 0.4, 0.5]$. Bold results indicate the best performing fraction per number of source tasks for each target task.

| | HalfCheetah | | | Hopper | | |
|---|---|---|---|---|---|---|
| $\lambda$ | 2 tasks | 3 tasks | 4 tasks | 2 tasks | 3 tasks | 4 tasks |
| 0.0 | 1841 ± 806 | 1752 ± 783 | 1742 ± 777 | 1917 ± 960 | 1585 ± 941 | 1263 ± 1113 |
| 0.1 | **2028 ± 993** | 2074 ± 762 | 1500 ± 781 | 2041 ± 887 | **6542 ± 4469** | 3300 ± 3342 |
| 0.2 | 1982 ± 838 | 1967 ± 862 | 1773 ± 748 | 1911 ± 712 | 5538 ± 4720 | 1702 ± 1078 |
| 0.3 | 1899 ± 911 | **2094 ± 859** | 1544 ± 771 | **2670 ± 1789** | 4925 ± 4695 | **3341 ± 4609** |
| 0.4 | 2008 ± 943 | 2015 ± 873 | **2162 ± 789** | 2076 ± 803 | 2437 ± 2731 | 1451 ± 991 |
| 0.5 | 1961 ± 944 | 1647 ± 896 | 1635 ± 809 | 1975 ± 772 | 2014 ± 2328 | 3246 ± 3828 |
| **Baseline** | | 1681 ± 726 | | | 1340 ± 1112 | |

Table 7: Average return for fraction transfer of 2, 3, and 4 source tasks for the Walker2D and Ant tasks with fractions $\lambda \in [0.0, 0.1, 0.2, 0.3, 0.4, 0.5]$. Bold results indicate the best performing fraction per number of source tasks for each target task.

| | Walker2D | | | Ant | | |
|---|---|---|---|---|---|---|
| $\lambda$ | 2 tasks | 3 tasks | 4 tasks | 2 tasks | 3 tasks | 4 tasks |
| 0.0 | 546 ± 776 | **545 ± 1012** | 157 ± 891 | 1070 ± 659 | 923 ± 559 | 897 ± 586 |
| 0.1 | 360 ± 803 | 441 ± 852 | **478 ± 1113** | 806 ± 469 | 1022 ± 645 | 834 ± 652 |
| 0.2 | **1009 ± 1254** | 393 ± 813 | 200 ± 807 | 1124 ± 722 | 1052 ± 687 | 898 ± 616 |
| 0.3 | 535 ± 914 | 325 ± 812 | 323 ± 818 | 1162 ± 824 | 1080 ± 701 | 905 ± 554 |
| 0.4 | 742 ± 992 | 516 ± 1074 | 354 ± 862 | 1308 ± 735 | 885 ± 571 | 1022 ± 709 |
| 0.5 | 352 ± 853 | 355 ± 903 | 396 ± 829 | 951 ± 651 | 932 ± 609 | 683 ± 498 |
| **Baseline** | | 116 ± 885 | | | **1589 ± 771** | |

Table 8: Average return for fraction transfer of 2, 3, and 4 source tasks for the InvertedPendulumSwingup and InvertedDoublePendulumSwingup tasks with fractions $\lambda \in [0.0, 0.1, 0.2, 0.3, 0.4, 0.5]$. Bold results indicate the best-performing fraction per number of source tasks for each target task.

| | InvertedPendulumSwingup | | | InvertedDoublePendulumSwingup | | |
|---|---|---|---|---|---|---|
| $\lambda$ | 2 tasks | 3 tasks | 4 tasks | 2 tasks | 3 tasks | 4 tasks |
| 0.0 | 847 ± 135 | 779 ± 259 | **838 ± 169** | 1255 ± 270 | 1303 ± 248 | 1208 ± 322 |
| 0.1 | 857 ± 145 | 834 ± 198 | 803 ± 239 | 1185 ± 316 | 1283 ± 272 | 1251 ± 272 |
| 0.2 | 856 ± 121 | **850 ± 139** | 782 ± 269 | 1209 ± 280 | 1299 ± 254 | 1235 ± 284 |
| 0.3 | **871 ± 93** | 832 ± 182 | 806 ± 213 | 1279 ± 216 | 1325 ± 225 | 1239 ± 259 |
| 0.4 | 858 ± 150 | 789 ± 285 | 790 ± 237 | **1307 ± 227** | 1323 ± 243 | **1278 ± 287** |
| 0.5 | 852 ± 126 | 830 ± 181 | 791 ± 304 | 1301 ± 216 | **1340 ± 230** | 1206 ± 278 |
| **Baseline** | | 723 ± 364 | | | 1194 ± 306 | |

## F   Algorithms

Algorithm 1 presents pseudocode of the original Dreamer algorithm (adapted from Hafner et al. (2020)) with seed episodes $S$, collect interval $C$, batch size $B$, sequence length $L$, imagination horizon $H$, and learning rate $\alpha$.

Algorithm 2 presents the adaptation of Dreamer such that it learns multiple tasks simultaneously. Finally, Algorithm 3 and Algorithm 4 present the procedures of fractional transfer learning using a multi-task agent and meta-model transfer learning for the reward model of Dreamer, respectively. Note that Algorithm 1 is the base algorithm, and we omit identical parts of the pseudocode in the other algorithms to avoid redundancy and enhance legibility.

---

**Algorithm 1**: Dreamer

---

Initialize dataset $\mathcal{D}$ with $S$ random seed episodes.
Initialize neural network parameters $\theta_{\text{REP}}, \theta_{\text{OBS}}, \theta_{\text{REW}}, \theta_{\text{TRANS}}, \phi, \psi$ randomly.
**while** *not converged* **do**

> **for** *update step c = 1 ... C* **do**
>
> > // Dynamics learning
> > Draw $B$ data sequences $\{(a_t, o_t, r_t)\}_{t=k}^{k+L} \sim \mathcal{D}$.
> > Compute model states $s_t \sim p_{\theta_{\text{REP}}}(s_t|s_{t-1}, a_{t-1}, o_t)$.
> > Update $\theta$ using representation learning.
> >
> > // Behavior learning
> > Imagine trajectories $\{(s_\tau, a_\tau)\}_{\tau=t}^{t+H}$ from each $s_t$.
> > Predict rewards $\mathbb{E}q_{\theta_{\text{REW}}}(r_\tau|s_\tau)$ and values $v_\psi(s_\tau)$.
> > Compute value estimates $\text{V}_\lambda(s_\tau)$.
> > Update $\phi \leftarrow \phi + \alpha\nabla_\phi \sum_{\tau=t}^{t+H} \text{V}_\lambda(s_\tau)$.
> > Update $\psi \leftarrow \psi\alpha\nabla_\psi \sum_{\tau=t}^{t+H} \frac{1}{2}\|v_\psi(s_\tau) - \text{V}_\lambda(s_\tau)\|^2$.
>
> // Environment interaction
> $o_1 \leftarrow$ `env.reset()`
> **for** *time step t = 1 ... T* **do**
>
> > Compute $s_t \sim p_{\theta_{\text{REP}}}(s_t|s_{t-1}, a_{t-1}, o_t)$ from history.
> > Compute $a_t \sim q_\phi(a_t|s_t)$ with the action model.
> > Add exploration noise to action.
> > $r_t, o_{t+1} \leftarrow$ `env.step($a_t$)`.
>
> Add experience to dataset $\mathcal{D} \leftarrow \mathcal{D} \cup \{(o_t, a_t, r_t)_{t=1}^{T}\}$.

---

---

**Algorithm 2**: Multi-Task Dreamer

---

Initialize dataset $\mathcal{D}$ with $S$ random seed episodes.
Initialize neural network parameters $\theta_{\text{REP}}, \theta_{\text{OBS}}, \theta_{\text{REW}}, \theta_{\text{TRANS}}, \phi, \psi$ randomly.
**while** *not converged* **do**
    `// Dynamics learning`
    ...
    `// Environment interaction`
    **for** *environment $i = 0 \dots N$* **do**
        $o_1 \leftarrow \text{env}_i.\texttt{reset()}$
        **for** *time step $t = 1 \dots T$* **do**
            Compute $s_t \sim p_\theta(s_t|s_{t-1}, a_{t-1}, o_t)$ from history.
            Compute $a_t \sim q_\phi(a_t|s_t)$ with the action model.
            Add exploration noise to action.
            $r_t, o_{t+1} \leftarrow \text{env}.\texttt{step}(a_t)$.
        Add experience to dataset $\mathcal{D} \leftarrow \mathcal{D} \cup \{(o_t, a_t, r_t)_{t=1}^T\}$.

---

**Algorithm 3**: Fractional Transfer Learning with Multi-Task Agent

---

`// Train multi-task agent`
$\theta_{\text{REP}_S}, \theta_{\text{OBS}_S}, \theta_{\text{REW}_S}, \theta_{\text{TRANS}_S}, \phi_S, \psi_S \leftarrow \text{train\_multi\_task}(\{\text{env}_0 \dots \text{env}_N\})$ (Algorithm 2)

`// Apply transfer learning`
Initialize neural network parameters $\theta_{\text{REP}}, \theta_{\text{OBS}}, \theta_{\text{REW}}, \theta_{\text{TRANS}}, \phi, \psi$ randomly.
$\theta_{\text{REP}} \leftarrow \theta_{\text{REP}_S}$
$\theta_{\text{OBS}} \leftarrow \theta_{\text{OBS}_S}$
$\theta_{\text{TRANS}} \leftarrow \theta_{\text{TRANS}_S}$
$\theta_{\text{REW}} \leftarrow \theta_{\text{REW}} + \lambda \theta_{\text{REW}_S}$
$\psi \leftarrow \psi + \lambda \psi_S$
$\phi \leftarrow \phi$
**while** *not converged* **do**
    **for** *update step $c = 1 \dots C$* **do**
        `// Dynamics learning`
        ...
        `// Behavior learning`
        ...
    `// Environment interaction`
    ...

---

**Algorithm 4**: Meta Model Transfer Learning for Reward Model

---

```
// Train multi-task agent to obtain UFS autoencoder
```
$\theta_{\text{REP}} \leftarrow$ train_multi_task($\{\text{env}_0 \dots \text{env}_N\}$) (Algorithm 2)

```
// Train individual source task agents
```
**for** $i = 0 \dots N$ **do**
$\quad \mid \quad m_{\theta_{\text{REW},i}} \leftarrow$ train($\text{env}_i$, $\theta_{\text{REP}}$) (Algorithm 1)

```
// Create meta-model and initialize parameters
```
$q_\Theta \leftarrow q_\Theta(\, r \mid s,\, q_{\theta_{\text{REW},i}}(r|s),\, \dots,\, q_{\theta_{\text{REW},N}}(r|s)\,)$
Initialize neural network parameters $\theta_{\text{OBS}}, \theta_{\text{TRANS}}, \phi, \psi$ randomly.
**while** *not converged* **do**
$\quad$ **for** *update step* $c = 1 \dots C$ **do**

$\quad\quad$ ```// Dynamics learning```
$\quad\quad$ ...

$\quad\quad$ ```// Behavior learning```
$\quad\quad$ Imagine trajectories $\{(s_\tau, a_\tau)\}_{\tau=t}^{t+H}$ from each $s_t$.
$\quad\quad$ Predict rewards $\mathbb{E}q_\Theta(r_\tau|s_\tau)$ and values $v_\psi(s_\tau)$.
$\quad\quad$ Compute value estimates $V_\lambda(s_\tau)$.
$\quad\quad$ Update $\phi \leftarrow \phi + \alpha \nabla_\phi \sum_{\tau=t}^{t+H} V_\lambda(s_\tau)$.
$\quad\quad$ Update $\psi \leftarrow \psi \alpha \nabla_\psi \sum_{\tau=t}^{t+H} \frac{1}{2}\|v_\psi(s_\tau) - V_\lambda(s_\tau)\|^2$.

$\quad$ ```// Environment interaction```
$\quad$ ...

---

