# OpenReview forum: "Multi-Source Transfer Learning for Deep Model-Based Reinforcement Learning"
_TMLR — Accepted by TMLR_

### Review · Reviewer_qCVq · 2023-02-28

**Summary Of Contributions:**

This paper proposes a model-based transfer framework to achieve multiple source policy transfer in the continuous control domain.

The contributions are summarized below:

1) A Fractional transfer learning mechanism is employed to transfer a part of parameters from source policies.

2) A meta-model transfer learning technique is proposed to transfer different components to the target task in a meta manner.

3) Experiments on cross-domain continuous control tasks show the effectiveness of the proposed method.

**Audience:**

Yes

**Broader Impact Concerns:**

Not applicable.

**Claims And Evidence:**

Yes

**Requested Changes:**

Major revisions should consider the literature review and experiment sections. Please refer to the pros and cons part.

Some inaccurate descriptions should also be clarified.

The typos should be fixed in the revised version.

Multiple entries for the same reference should be fixed.

**Strengths And Weaknesses:**

Strengths:

The proposed method is novel from the perspective of model-based transfer learning.

The paper is clearly written and technically substantiated.

Weaknesses:

The literature review is not extensive, especially missing the discussion about cross-domain transfer methods, such as [1,2].

Furthermore, I am not convinced with the statement 'to the best of our knowledge, we are the first to demonstrate the application of this type of transfer learning in deep RL settings.' It has been used for several years, since [3]. Two different ways of fraction transfer learning: one is a weighted combination of network layer output [1-3], and the other is the weighted combination of network layer weights [4,5].

No other transfer baselines are covered in the experiment section. It's better to provide SOTA baselines, such as [6], (rather than variants of the proposed methods, this is more like ablation studies), or explain why other baselines are not comparable.

The structure of Sections 5 and 6 could be reorganized and optimized. There is little discussion about the training results in Section 5.

[1] Cross-domain Adaptive Transfer Reinforcement Learning Based on State-Action Correspondence

[2] Mutual Information Based Knowledge Transfer Under State-Action Dimension Mismatch.

[3] Progressive Neural Networks.

[4] Knowledge Flow: Improve Upon Your Teachers.

[5] Zoo-Tuning: Adaptive Transfer from a Zoo of Models.

[6] EUCLID: Towards Efficient Unsupervised Reinforcement Learning with Multi-choice Dynamics Model.

---

> ### Author Response · Authors · 2023-03-06
> **Response to Reviewer qCVq**
>
> Thank you for your feedback!
>
> > "The literature review is not extensive, especially missing the discussion about cross-domain transfer methods, such as [1,2]. Furthermore, I am not convinced with the statement 'to the best of our knowledge, we are the first to demonstrate the application of this type of transfer learning in deep RL settings.' It has been used for several years, since [3]. Two different ways of fraction transfer learning: one is a weighted combination of network layer output [1-3], and the other is the weighted combination of network layer weights [4,5].."
>
> Thank you for these references, we will make sure to incorporate them into the paper! We also thank you for pointing out we are not the first to demonstrate the application of FTL-type transfer learning in deep RL, we instead intended to say we are the first to adapt it to the model-based deep RL application, which we will correct in the paper.
>
> > "No other transfer baselines are covered in the experiment section. It's better to provide SOTA baselines, such as [6], (rather than variants of the proposed methods, this is more like ablation studies), or explain why other baselines are not comparable."
>
> Unfortunately, we are not aware of comparable transfer learning alternatives that are applicable to cross-domain multi-source settings and complex world model-based algorithms that also have code publicly available. For instance, [6] is designed for transfer learning in a very different setting:
>  - They focus on unsupervised dynamics model pre-training, where merely the dynamics model is pre-trained trained without any reward function or policy learning.
> - Policy and reward-related components are not transferred at all.
> - The transfer learning happens in a single-source manner, and more importantly, within a single domain (e.g. from Walker to Walker) with different reward functions, which is a much easier problem than multi-source cross-domain transfer learning as our methods are addressing.
>
> Similarly, although [1] and [2] also attempt to tackle multi-source cross-domain transfer learning, they specifically focus on policies and simple model-free algorithms, meaning these are not compatible with world model-based architectures. Note that our framework is difficult to compare with such methods, as our techniques are focused on the model-based components (dynamics model, autoencoder, reward model).
>
> We will make sure to emphasize in the paper why other baselines are not comparable, and we appreciate that you pointed this out!
>
> In the general response to all of the reviewers you can find the details of the adaptations we made to the paper based on your and the other reviewers' feedback. We hope to have addressed your concerns in this response and with the adaptations of the paper, and would greatly appreciate it if you could tell us whether they have resolved your concerns!

---

### Review · Reviewer_eue6 · 2023-03-03

**Summary Of Contributions:**

This paper addresses the challenges in multi-source transfer learning for deep reinforcement learning problems with the following techniques developed upon an existing model-based continuous control framework Dreamer: (1) **fractional transfer learning (FTL)**, where the target task initialization is defined as a weighted sum between random initialized values plus scaled source parameter values; (2) **modular transfer learning**, where among different components for Dreamer, some of them should be *fully transferred* (e.g., representation, observation, and transition) whereas the others should be *fractionally transferred* (e.g., reward function and value function); (3)  **universal feature space** where the encoder of a trained autoencoder can be reused across tasks can be used together with a **meta-model transfer learning (MMTL)** module, where the later one ensembles the latent prediction from all source task models to be concatenated to the latent of target task, to let agent automatically learn to distinguish relevant and irrelevant features.

For empirical evaluation, the authors consider six continuous control domains as the target task, respectively, where the multi-task transfer adopts source domains with sizes 2 to 4, with the source domain consisting of both positively and negatively correlated domains to the target task. The authors evaluate FTL, MMTL, and Dreamer (baseline) in multi-task learning settings.

**Audience:**

Yes

**Claims And Evidence:**

No

**Requested Changes:**

- **Problems with references**: (1) many of the citations are out of date, e.g., "Distral: Robust multitask reinforcement learning" and "Efficient deep reinforcement learning via adaptive policy transfer" are all published as conference papers, but both are cited as arxivs;  (2) some arxiv citations come with URL but the others not; (3) the format for all conference proceedings and  should be aligned properly.

- **Typo**:  (page 4) we propose to leverage parameter fusion -> add a full stop at the end of the sentence.

- The appearance of Table 1 and Table 2 could be moved before Section 6.

- It would be good if an illustrative figure on Dreamer and Dreamer's important math formulas could be added to Section 2 when the authors attempt to introduce Dreamer in preliminary.

- Wrap Figure 1 and Figure 2 with proper margins.

- Table 2 caption: 1e-5 should be changed to 1e5?

- Redraw Figure 2 and Figure 2 with a clear relationship between source tasks and the target task.

- It would be good if algorithms will be added to either the main body or the appendix to describe the procedure of MTL with the proposed methods.

Please refer to the weaknesses part for other suggestions on the formulation, motivation, and empirical evaluation.

**Strengths And Weaknesses:**

**Strengths**:
- This paper learns an important yet less studied problem of multi-source policy transfer in deep reinforcement learning.
- The paper motivates a novel angle that considers a model-based approach to address the challenge in multi-source transfer learning, while most conventional works fall into the category of model-free policy learning.

**Weaknesses**:
- This paper highly depends on an existing model-based continuous control method Dreamer, which reduces both the technical novelty and generality of the proposed method. It is good to think from the model-based perspective, but regretfully the authors did not manage to develop solid technical solutions to address model-based multi-source policy transfer. The only thing that seems related is the fractional parameterization part, where the authors put a very strong assumption that reward and value modules are highly encouraged to apply FTL whereas the other modules in Dreamer, like representation and transition, need full transfer. The idea of applying FTL to model-based Dreamer appears to be a bit heuristic and trivial, with missing empirical evidence. I also feel it is very hard to apply FTL or the meta-model transfer MMTL to other model-based solutions apart from Dreamer to solve general multi-source transfer problems.

- The formulation of the method is not very sound, with many important concepts about the multi-source transfer problem left undefined. I think the problem tackled by this paper falls into one of the multi-source transfer learning categories, where with pre-trained source task knowledge/models, their method adds *one* target task to the source task set and updates those tasks thereafter to train the target task. The aforementioned type is not the only way multi-source transfer can be formed. For instance, earlier DQN-variant of multi-source policy transfer methods such as policy distillation [1] and AMN [2] trains multiple target domains simultaneously. I feel the problem setting could be more clearly defined. Some other important concepts, such as *full transfer* and *direct transfer* that has been mentioned several times in the paper are also undefined.

- There are very few math formulas in the method section, which makes the writing hard to follow. Some important formulas are ambiguously defined, e.g., FTL update is defined as $\theta_T \leftarrow \theta_\epsilon + \lambda \theta_S$ in the paper, which I think is wrong, as $\theta_S$ is actually a parameter set. Is $\theta_S$ frozen during training? Also, for FTL, the parameters over multiple source domains are straightforwardly added to a random initialization, weighted by a predefined scalar value, which is a bit heuristic. It would be more interesting if the method could automatically adjust $\lambda$ by itself.

- I'm not convinced about the feature concatenation idea for the meta-model transfer learning (MMTL) part. First, the authors require source tasks to inference state from the disjoint target task to generate the latent feature to be ensembled, which I feel might not be valid for general multi-source transfer learning problems. Second, it is hard to define which layer of the neural network is most suitable to extract the additional feature in MMTL. Last but not least, the authors make a very strong claim that their method could learn the most relevant source task, for which I disagree. It is unclear what kind of weight the inference module would give over each source task embedding from $m_{\theta_0}(y|x), ..., m_{\theta_N}(y|x)$, i.e., if source task 0 and N are positively and negatively correlated with the target task, would the MMTL model always give higher weight to $m_{\theta_0}(y|x)$ than $m_{\theta_N}(y|x)$?

- The empirical evaluation results are interesting. I think the results try to tell us that if we train a target task in a single-task way, we would result in the scores under the 'Baseline' (Dreamer)'s standard; but if we use more tasks (target + multiple source tasks), with known knowledge on the source tasks, plus the FTL/MMTL technique proposed in this paper, we can achieve better score *for the target task* after the multitask training. One baseline is necessary to add, which is directly training the target task with source tasks together without FTL or MMTL. I would also be curious about how much additional computational complexity would the transfer learning enquire compared to training from scratch. It would also be good if the authors could present results on fully transferring or not each of the modules in Dreamer.

- It's common sense that we could not group random tasks together for multi-task transfer learning, and it is essential to know what kind of problems can be effectively tackled by the method presented in this paper. In many places, the authors have mentioned terms like *task similarity* and *visual similarity* to measure the relationship between tasks, for which I think a definition is needed. It is also good to know apart from the six continuous control problems, what are the problem domains that the proposed method could address.

- For future work, the authors have mentioned an extremely interesting direction, which is to apply the method to Atari 2600 domain. Since there is a version of Dreamer for Atari, could the authors comment on whether the method could successfully work on multi-source transfer learning with Atari 2600 games, where the input of the method is high-dimensional and comes with visual differences at game-specific pixel-level?

[1] POLICY DISTILLATION.

[2] ACTOR-MIMIC DEEP MULTITASK AND TRANSFER REINFORCEMENT LEARNING.

---

> ### Author Response · Authors · 2023-03-06
> **Response to Reviewer eue6 1/2**
>
> Thank you for your feedback!
>
> We appreciate the requested changes and will make sure to incorporate them in the revision of the paper. It is not quite clear to us whether the items listed in the requested changes are most important for you to consider acceptance, or that all weaknesses would need to be addressed accordingly as well. It would be great if you could elaborate on that matter.
>
> We will take your requested changes into account along with suggestions on clarity in the weaknesses. We will attempt to address and clarify your concerns below:
>
> > "This paper highly depends ... multi-source policy transfer."
>
> As motivated in the introduction, we focus on world model-based algorithms as they achieve state-of-the-art sample efficiency in many benchmarks. Therefore, we have researched transfer learning techniques that are compatible with such complex architectures that learn several components within a latent space, meaning Dreamer is the most logical choice for evaluation. Due to limited computational resources, we decided to focus on the state-of-the-art model-based algorithm as we believe there is more value in strong empirical evidence for one algorithm than weak empirical evidence for multiple algorithms. You are also concerned with not addressing model-based multi-source policy transfer, which is a very challenging problem in cross-domain applications as the environments have different state-action spaces, meaning such policy transfer only really makes sense within the same domain or very similar domains. Instead, our transfer learning techniques result in enhanced learning efficiency of a new policy by facilitating significantly faster learning of the model-based components such as the representation model, dynamics model, and reward model, and also the value model for the policy.
>
> > "The idea of applying FTL to model-based Dreamer appears to be a bit heuristic and trivial, with missing empirical evidence."
>
> The motivation behind FTL in this setting is that randomly initializing reward and value functions means we discard potentially useful information. Instead, by preserving a portion of the source task parameters, we can transfer useful information without overfitting the parameters: see Appendix E for exhaustive empirical results and evidence for FTL.
>
> > "I also feel ... multi-source transfer problems".
>
> MMTL and FTL are applied on a modular level, meaning they should be applicable to any component of any algorithm. However, our paper focuses on multi-source settings to avoid the manual selection of an optimal source task. The multi-task setting requires an algorithm that can learn multiple tasks simultaneously. As was shown in [1], the Dreamer line of algorithms is able to learn multiple tasks simultaneously, whereas this is not necessarily known for other model-based algorithms.
>
> >"There are very few math formulas in the method section ... could automatically adjust by itself."
>
> To clarify, the formula you refer to is not the update of the parameters, but the initialization of the parameters. That is, before training, the parameters of the target agent are initialized as the randomly initialized weights summed with a fraction of the source agent's weights. This should also clarify why automatically learning $\lambda$ is unfortunately not an option, as it is a hyperparameter set before training rather than a parameter that can be adjusted during training: in Appendix E an extensive hyperparameter sweep for $\lambda$ can be found.

---

> ### Author Response · Authors · 2023-03-09
> **Response to Reviewer eue6 2/2**
>
> >I'm not convinced  ...  give higher weight ...?
>
> Note that we are not concatenating features but outputs of the reward models, being the predicted mean of a Gaussian for the reward model of Dreamer, we just realized this is wrongly described in Figure 2 but is explained correctly in 4.2. Additionally, note that we do not claim the method learns the most relevant source task but is able to extract the most relevant information from the pool of source tasks. This could also be information from multiple sources, as multiple environments may be relevant for a given target environment. Therefore MMTL may weigh multiple sources positively or negatively.
>
> > The empirical evaluation results are interesting ... or not each of the modules in Dreamer.
>
> We do in fact make the comparisons you suggest in Appendix D, where we compare the performance of finetuning a pre-trained Dreamer agent without FTL or MMTL (i.e. with full transfer). As can be seen, this results in detrimental performances, and therefore the baseline was used that learns from scratch. Moreover, Appendix E shows results for omitting FTL as well ($\lambda = 0$). Note that for FTL there is no additional computational complexity introduced. Even though there is additional inference of N frozen reward model, for MMTL there is also no noticeable additional computational complexity as the frozen autoencoder does not require gradient updates, compensating for any additional computation.
>
> >For future work, the authors ... game-specific pixel-level?
>
> This is indeed an extremely interesting question that can only be confidently answered with empirical evidence. However, we believe that the methods could provide for improved performances. The learned convolutional features of the autoencoder on multiple Atari games should provide better sample efficiency compared to learning from scratch, as we expect some generalization to the new game.  If a game such as Pong is included in the source set, and a dynamically similar game such as Breakout is the target task, we would naturally expect better performance improvements compared to less related games. In this scenario, it would be appropriate to transfer the dynamics model with FTL or MMTL given the similarities in transition function. However, note that in Atari games the reward function is very different across games in contrast to the locomotion/pendula tasks, meaning that the transfer of the reward model may be more challenging. In contrast, the value model would be better suited for transfer (e.g. with FTL) as valuating hitting/missing a ball with a paddle in Pong may generalize to the same act in Breakout.
>
> In the general response to all of the reviewers you can find the details of the adaptations we made to the paper based on your and the other reviewers' feedback. We hope to have addressed your concerns in this response and with the adaptations of the paper, and would greatly appreciate it if you could tell us whether they have resolved your concerns!

---

### Review · Reviewer_YJjx · 2023-03-06

**Summary Of Contributions:**

This paper presents multiple transfer learning techniques to extend the model based reinforcement learning algorithm dreamer. The authors discuss several methods for transfer learning, one where a single agents interacts with multiple environments and one where the knowledge from multiple agents is used to solve a target environment.
The authors then evaluate the proposed transfer learning methods on continuous control problem and show that it improves the vanilla Dreamer algorithm.

**Audience:**

Yes

**Broader Impact Concerns:**

No broader impact concerns.

**Claims And Evidence:**

No

**Requested Changes:**

Some changes I would require for acceptance:

- "We propose techniques for two different multi-source transfer learning settings: a single agent that has mastered multiple tasks, and multiple individual agents that each mastered a single task" I think it would be better to focus on a single setting and provide a more thorough investigation of the proposed methods in this setting.
-The paper should update its section on transfer learning in deep rl and include stronger baselines to compare with their algorithm.
- Show that the proposed techniques are not limited to Dreamer and could be applied with success on another model based learning algorithm.

Nice to see:

- Experiments on environments using pixels as inputs like Atari or Mujoco from pixels.

**Strengths And Weaknesses:**

Pros:

- The paper studies a key problem in reinforcement learning, how to reuse prior/previously learned knowledge is currently an active area of research.

Cons:

- Overall I found the paper really hard to follow. The paper mentions many techniques in two different setups, it is hard to contribution of each of these techniques. Many terms are introduced while they already exists in the literature and the definition are not properly highlighted in the document. In the end I had a hard time to differentiate simultaneous multi task learning, meta model transfer learning, modular transfer learning while the paper barely has space to investigate the quality of the proposed techniques.
- The paper mentions many algorithms / techniques but do not provide a pseudo code of the final algorithm, making it hard to understand the actual algorithm evaluated. It would also be useful to provide an ablation of the different components of the proposed algorithms to understand their relative importance.
- The paper ignores a lot of existing literature on transfer learning in reinforcement learning and even mentions that: "transfer learning receives little attention in RL research.", "The transferability of deep model-free RL algorithms doesn’t appear promising". I refer the authors to the call of papers of the upcoming workshop on Reincarnating RL that includes many relevant references: The transferability of deep model-free RL algorithms doesn’t appear promising.
- The paper focuses on Dreamer, does it mean that the proposed technique only works with this algorithm or it would work with any model based reinforcement learning algorithm? The paper also does not justify why it focuses on model based algorithms.
- The baseline used in the paper is a Dreamer algorithm trained from scratch, this is a weak baseline in the context of transfer learning, simply finetuning the weights pretrained on the source task would be a stronger baseline (though I recommend the authors to compare with existing transfer learning algorithm).
- "we are the first to propose a solution that allows the combination of models from agents trained in different environments, from which the most relevant in- formation can autonomously be extracted for a given target task." this has been done previously e.g progressive networks. The relationship between Meta model transfer learning and progressive was actually highlighted by authors. Though it might be worth mentioning that this technique is not widely used due the fact that computation grows linearly with the number of tasks.

---

> ### Author Response · Authors · 2023-03-06
> **Response to Reviewer YJjx**
>
> Thank you for your feedback! We will take the cons into account for revision such as improving the related works section and providing clearer presentations of the methods, and would like to discuss the requested changes below :
>
> >"We propose techniques ... in this setting."
>
> We believe there is some confusion here concerning the purpose of each of the proposed methods, as the two proposed techniques are each solutions for specifically one of the two settings: MMTL allows multi-source transfer of multiple agents, whereas the simultaneous multi-task learning combined with FTL allows multi-source transfer in terms of a multi-task agent. Therefore, we cannot omit one of the settings from the paper. Moreover, we believe it is an important contribution to have solutions to multiple multi-source settings that researchers can face.
>
> >"The paper should update its section on transfer learning in deep rl" "
>
> We will update our related works section as you and the other reviewers suggest, thank you for pointing this out. Regarding our claim that model-free algorithms don't appear promising for transfer learning, note that we are considering transfer across transfer and domains in this statement. The framework you are referring to with Reincarnating RL is a policy distillation framework, that only focuses on transfer within the same domain and the same task. As we mention in the paper in the next sentence, although model-free algorithms don't appear promising for cross-domain/task transfer, they are suitable for distillation techniques. However, we agree that this should be further clarified in the paper and will make sure that we also mention recent more successful model-free applications.
>
> > "and include stronger baselines to compare with their algorithm."
>
> You suggest comparing the performance to a fine-tuned agent, rather than training from scratch. As can be seen in Appendix D, simply fine-tuning the parameters resulted in negative transfers compared to training from scratch, which both shows why our approach is crucially important and why we chose to compare the performance with training an agent from scratch. Unfortunately, there are no comparable transfer learning alternatives that are applicable to cross-domain multi-source settings and complex world model-based algorithms that also have code publicly available.  We would greatly appreciate it if the reviewer is aware of suggestions that fit these criteria. However, we do agree that it would be extremely interesting to apply these techniques to other algorithms and settings, as we propose in the future work paragraph of the conclusion.
>
> >"Show that the proposed techniques are not limited to Dreamer and could be applied with success on another model based learning algorithm."
>
> The reasons for applying the proposed techniques solely to Dreamer algorithms are:
> - As motivated in the introduction, we focus on world model-based algorithms as they achieve state-of-the-art sample efficiency in many benchmarks. Therefore, we have researched transfer learning techniques that are compatible with such complex architectures that learn several components within a latent space, meaning Dreamer is the most logical choice for evaluation.
> -  Due to limited computational resources, we decided to focus on the state-of-the-art model-based algorithm as we believe there is more value in strong empirical evidence for one algorithm than weak empirical evidence for multiple algorithms.
> - MMTL and FTL are applied on a modular level, meaning they should be applicable to any component of any algorithm. However, our paper focuses on multi-source settings to avoid the manual selection of an optimal source task. The multi-task setting requires an algorithm that can learn multiple tasks simultaneously. As was shown in [1], the Dreamer line of algorithms is able to learn multiple tasks simultaneously, whereas this is not necessarily known for other model-based algorithms.
>
> However, it would indeed be extremely interesting to apply these techniques to other algorithms and settings, as we propose in the future work paragraph of the conclusion.
>
> > "Experiments on environments using pixels as inputs like Atari or Mujoco from pixels."
>
> All experiments in this paper were done using pixel inputs: note that Dreamer is specifically designed with a convolutional autoencoder that maps images to latent states.
>
> In the general response to all of the reviewers you can find the details of the adaptations we made to the paper based on your and the other reviewers' feedback. We hope to have addressed your concerns in this response and with the adaptations of the paper, and would greatly appreciate it if you could tell us whether they have resolved your concerns!
>
> [1] Learning Latent Dynamics for Planning from Pixels

---

### Decision · Action_Editors · 2023-04-26

**Recommendation:** Accept as is

**Comment:**

This paper deals with multi-source transfer learning for dreamer, as an example of (deep) model-based RL. The initial reviews raised some concerns, such as clarity or links to progressive networks. The rebuttal and revision adequately addressed these concerns, I therefore follow the reviewers in recommending acceptance.

**Audience:**

Yes

**Claims And Evidence:**

Yes